# Vispi: Automatic Visual Perception and Interpretation of Chest X-rays

**Xin Li** [1]                                                                                                           XINLEE@WAYNE.EDU
**Rui Cao** [2]                                                                            CAORUI@STUMAIL.NWU.EDU.CN
**Dongxiao Zhu**[1]                                                                                     DZHU@WAYNE.EDU

[1] *Department of Computer Science, Wayne State University, USA*

[2] *School of Information Science and Technology, Northwest University, China*

## Abstract

Medical imaging contains the essential information for rendering diagnostic and treatment decisions. Inspecting (visual perception) and interpreting image to generate a report are tedious clinical routines for a radiologist where automation is expected to greatly reduce the workload. Despite rapid development of natural image captioning, computer-aided medical image visual perception and interpretation remain a challenging task, largely due to the lack of high-quality annotated image-report pairs and tailor-made generative models for sufficient extraction and exploitation of localized semantic features, particularly those associated with abnormalities. To tackle these challenges, we present Vispi, an automatic medical image interpretation system, which first annotates an image via classifying and localizing common thoracic diseases with visual support and then followed by report generation from an attentive LSTM model. Analyzing an open IU X-ray dataset, we demonstrate a superior performance of Vispi in disease classification, localization and report generation using automatic performance evaluation metrics ROUGE and CIDEr.

**Keywords:** Medical Image Report Generation, Disease Classification and Localization, Visual Perception, Attention, Deep Learning

## 1. Introduction

X-ray is a widely used medical imaging technique in clinics for diagnosis and treatment of thoracic diseases. Medical image interpretation, including both disease annotation and report writing, is a laborious routine for radiologists. Moreover, the quality of interpretation is often quite diverse due to the differential levels of experience, expertise and workload of the radiologists. To release radiologists from their excessive workload and to better control quality of the written reports, it is desirable to implement a medical image interpretation system that automates the visual perception and cognition process and generates draft reports for radiologists to review, revise and finalize.

Despite the rapid and significant development, the existing natural image captioning models, e.g. Krause et al. (2017); Xu et al. (2015), fail to perform satisfactorily on medical report generation. The major challenge lies in the limited number of image-report pairs and relative scarcity of abnormal pairs for model training, which are essential for quality radiology report generation. Additional challenge is the lack of appropriate performance evaluation metrics; the $n$-gram based BLEU scores widely used in natural language processing (NLP) are not suitable for assessing the quality of generated reports.

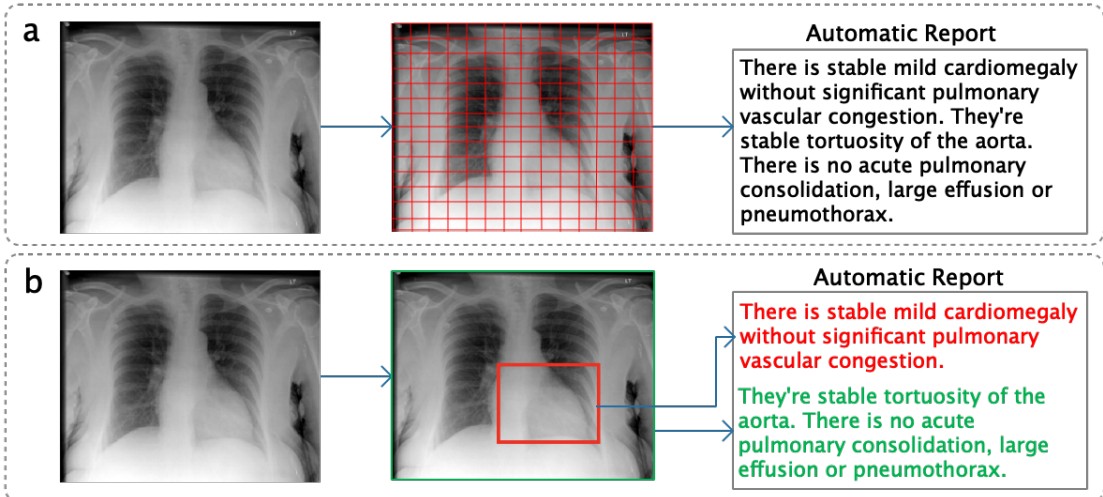

Figure 1: Illustration of an existing medical report generation system (e.g. Jing et al. (2017); Xue et al. (2018)) (a) and the proposed medical image interpretation system (b). The former uses a coarse grid of image regions as visual features to generate report directly whereas the latter first predicts and localizes disease as semantic features then followed by report generation.

Nevertheless several approaches have been developed to generate reports automatically for chest X-rays using the CNN-RNN architecture developed in natural image captioning research (Jing et al., 2017; Li et al., 2018; Wang et al., 2018; Xue et al., 2018) (Fig. 1a). Since the medical report typically consists of a sequences of sentences, Jing et al. (2017) use a hierarchical LSTM (Krause et al., 2017) to generate paragraphs and achieve impressive results on Indiana University (IU) X-ray dataset (Demner-Fushman et al., 2015). Instead of only using visual features extracted from image, they first predict the Medical Text Indexer (MTI) annotated tags, and then combine semantic features from the tags with visual features from the images for report generation. Similarly, Xue et al. (2018) use both visual and semantic features but generate 'impression' and 'findings' of the report separately. The former one-sentence summary is generated from a CNN encoder whereas the latter paragraph is generated using visual and semantic features. Different from CoAtt, the semantic feature is extracted by embedding the last generated sentence as opposed to the annotated tags. Li et al. (2018) use a hierarchical decision-making procedure to determine whether to retrieve a template sentence from an existing template corpus or to invoke the lower-level decision to generate a new sentence from scratch. The decision priority is updated via reinforcement learning based on sentence-level and word-level rewards or punishments. However, none of these methods demonstrate a satisfactory performance in disease localization and classification, which is a central issue in medical image interpretation.

TieNet (Wang et al., 2018) address both disease classification and medical image report generation problems in the same model. They introduce a novel Text-Image Embedding network (TieNet), which integrates self-attention LSTM using textual report data and visual

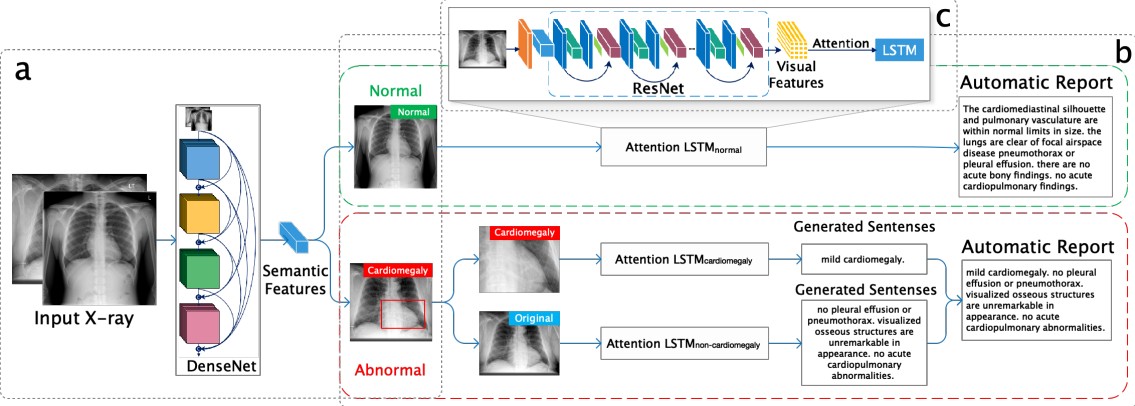

Figure 2: An automatic workflow of the X-ray interpretation system.

attention CNN using image data. TieNet is capable of extracting an informative embedding to represent the paired medical image and report, which significantly improves the disease classification performance compared to Wang et al. (2017). However, TieNet's performance on medical report generation improves only marginally over the baseline approach (Xu et al., 2015), trading the medical report generation performance for the disease classification performance. Moreover, TieNet does not provide a visual support for radiologists to review and revise the automatically generated report.

We present an automatic medical image interpretation system with *in situ* visual support striving for a better performance in both image annotation and report generation (Fig. 1b). To our knowledge this is among the first attempts to exploit disease localization for X-ray image report generation with visual supports. Our contributions are in four-fold: (1) we describe an integrated image interpretation framework for disease annotation and medical report generation, (2) we transfer knowledge from large image data sets (ImageNet and ChestX-ray8) (Wang et al., 2017) to enhance medical image interpretation using a small number of reports for training (IU X-ray) (Demner-Fushman et al., 2015), (3) we evaluate suitability of the NLP evaluation metrics for medical report generation, and (4) we demonstrate the functionality of localizing the key finding in an X-ray with a heatmap.

## 2. Method

Our workflow (Fig. 2) first annotates an X-ray image by classifying and localizing thoracic diseases (Fig. 2a) and then generates the corresponding sentences to build up the entire report (Fig. 2b). Fig. 2c displays the structure of attentive LSTM used to generate reports.

### 2.1. Disease Classification and Localization

Fig. 2a shows our classification module built on a 121-layer Dense Convolutional Network (DenseNet) (Huang et al., 2017). Similar to Rajpurkar et al. (2017), we replace the last fully-connected layer with a new layer of dimension $M$, where $M$ is the number of diseases. This is a multiple binary classification problem that input is a frontal view X-ray image

$\mathbf{X}$ and output is a binary vector $\mathbf{y} = [y_1, \ldots, y_m, \ldots, y_M]$, i.e., $y_m \in \{0, 1\}$, indicating absence or presence of a disease $m$. The binary cross-entropy loss function is defined as: $L(\mathbf{X}, \mathbf{y}) = -\sum_{m=1}^{M}[y_m(\log g_m(\mathbf{X})) + (1 - y_m)\log(1 - g_m(\mathbf{X}))]$, where $g_m(\mathbf{X})$ is the probability for a target disease $m$. If $g_m(\mathbf{X}) > 0.8$, an X-ray is annotated with disease $m$ for the next level modeling. Otherwise, it is considered as "Normal". It is worth mentioning that a vast majority of X-rays are considered as "Normal", therefore, other choices of thresholds also work well with our system.

We apply Grad-GAMs (Selvaraju et al., 2017) to localize disease with a heatmap. Gard-CAMs uses the gradient information and flows it back to the final convolutional layer to decipher the importance of each neuron in classifying an image to disease $m$. Formally, let $\mathbf{A}_k$ be the $k$th feature maps and weight $w_{mk}$ represents importance of the feature map $k$ for the disease $m$. We first calculate the gradient of the score for class m, $z_m$ (before the sigmoid), with respect to a feature map $\mathbf{A}_k$, i.e., $\frac{\partial z_m}{\partial \mathbf{A}_k}$. Thus $w_{mk}$ are calculated by: $w_{mk} = \frac{1}{N}\sum_i\sum_j \frac{\partial z_m}{\partial \mathbf{A}_k}$. $(i, j)$ represents the coordinates of a pixel, and $N$ is the total number of pixels. We then generate a heatmap for disease $m$ by applying weighted average of $\mathbf{A}_k$, followed by a ReLU activation: $\mathbf{H}_m = \text{ReLU}(\sum_k w_{mk}\mathbf{A}_k)$. The localized semantic features to predict disease $m$ are identified and visualized with the heatmap $\mathbf{H}_m$. Similar to Wang et al. (2017), we apply a thresholding based bounding box (B-Box) generation method. The B-Box bounds pixels whose heatmap intensity is above 90% of the maximum intensity. The resulting region of interest is then cropped for next level modeling.

## 2.2. Attention-based Report Generation

Fig. 2b illustrates the process of report generation. If there is no active thoracic disease found in an X-ray, a report will be directly generated by an attentive LSTM based on the original X-ray as shown in the green dashed box. Otherwise (as shown in the red dashed box), the cropped subimage with localized disease from the classification module (Fig. 2a) is used to generate description of abnormalities whereas the original X-ray is used to generate description of normalities in the report.

As shown in the Fig. 2c, the attentive LSTM is based on an encoder-decoder structure (Xu et al., 2015), which takes either the original X-ray image or the cropped subimage corresponding to abnormal region as the input and generates a sequence of sentences for the entire report. Our encoder is built on a pre-trained ResNet-101 (He et al., 2016), which extracts the visual features matrix $\mathbf{F} \in \mathbb{R}^{2048 \times 196}$ (reshaped from $2048 \times 14 \times 14$) from the last convolutional layer followed by an adaptive average pooling layer. Each vector $\mathbf{F}_k \in \mathbb{R}^{2048}$ of $\mathbf{F}$ represents one regional feature vector, where $k = \{1, ..., 196\}$.

The LSTM decoder takes $\mathbf{F}$ as input and generates sentences by producing a word $\mathbf{w}_t$ at each time $t$. To utilize the spatial visual attention information, we define the weights $\alpha_{tk}$, which can be interpreted as the relative importance of region feature $\mathbf{F}_k$ at time $t$. The weights $\alpha_{tk}$ is computed by a multilayer perceptron $f$: $e_{tk} = f(\mathbf{F}_k, \mathbf{h}_{t-1})$ and $\alpha_{tk} = \text{Softmax}(e_{tk})$, and hence the attentive visual feature vector $\mathbf{V}_t$ is computed by $\mathbf{V}_t = \sum_{k=1}^{196} \alpha_{tk}\mathbf{F}_k$. In addition to the weighted visual feature $\mathbf{V}_t$ and last hidden layer $\mathbf{h}_{t-1}$, the RNN also accepts the last output word $\mathbf{w}_t$ at each time step as an input. We concatenate the embedding of last output word and visual feature as context vector $\mathbf{c}_t$. Thus the transition to the current hidden layer $\mathbf{h}_t$ can be calculated as: $\mathbf{h}_t = \text{LSTM}(\mathbf{c}_t, \mathbf{h}_{t-1})$.

After model training, a report is generated by sampling words $\mathbf{w}_t \sim p(\mathbf{w}_t|\mathbf{h}_t)$ and updating the hidden layer until hitting the stop token.

## 3. Experiments and Results

**Datasets.** We use the IU Chest X-ray Collection (Demner-Fushman et al., 2015), an open image dataset with 3955 radiology reports paired with chest X-rays (one study per patient) for our experimental evaluation. Each report contains three sections: impression, findings and Medical Subject Headings (MeSH) terms. Similar to Jing et al. (2017); Xue et al. (2018), we generate sentences in 'impression' and 'findings' together. The MeSH terms are used as labels for disease classification (Wang et al., 2018) as well as the follow-up report generation with abnormality and normality descriptions. We convert all the words to lower-case, remove all non-alphanumeric tokens, replace single-occurrence tokens with a special token and use another special token to separate sentences. We filter out images and reports that are non-relevant to the eight common thoracic diseases included in both ChestX-ray8 (Wang et al., 2017) and IU X-ray datasets (Demner-Fushman et al., 2015), resulting in a dataset with 2225 pairs of X-ray image and report. Finally, we split all the image-report pairs into training, validation and testing dataset by ratio 7 : 1 : 2 without patient overlap.

**Implementation Details.** We implement our model on a GeForce GTX 1080ti GPU platform using PyTorch. The dimension of all hidden layers and word embeddings are set to 512. The network is trained with Adam optimizer with a mini-batch size of 16. The training stops when the performance on validation dataset does not increase for 20 epochs. We do not fine-tune the DenseNet pretrained with ChestX-ray8 (Wang et al., 2017) and ResNet pretrained with ImageNet due to the small sample size of IU X-ray dataset (Demner-Fushman et al., 2015). For each disease class, a specific pair of LSTMs are trained to ensure consistency between the predicted disease annotation(s) and the generated report. For the disease classes with less than 50 samples, we train a shared attentive LSTM across the classes to generate normality description of the report.

**Evaluation of Automatic Medical Image Reports.** We use the metrics for NLP tasks such as BLEU (Papineni et al., 2002), ROUGE (Lin, 2004), and CIDEr (Agrawal et al., 2017) for automatic performance evaluation. As shown in Table 1, our model outperforms all baseline models (Donahue et al., 2015; Lu et al., 2017; Rennie et al., 2017; Vinyals et al., 2015) and demonstrates the best CIDEr and ROUGE scores among all the advanced methods specifically designed for medical report generation (Jing et al., 2017; Li et al., 2018; Xue et al., 2018), despite the fact that we only use a single frontal view X-ray. While BLEU scores measure the percentage of consistency between the automatic report and the manual report in light of the automatic report (precision), it is not illuminative in assessing the amount of information captured in the automatic report in light of the manual report (recall). In real-world clinical applications, both recall and precision are critical in evaluating the quality of an automatic report.

For example, automatic reports often miss description of abnormalities that contained in manual reports written by human radiologists (Li et al., 2018; Xue et al., 2018), which may decreases recall but does not affect precision. Thus, the automatic report missing the

| Model | CIDEr | ROUGE | BLEU-1 | BLEU-2 | BLEU-3 | BLEU-4 |
|---|---|---|---|---|---|---|
| CNN-RNN (Vinyals et al., 2015)* | 0.294 | 0.306 | 0.216 | 0.124 | 0.087 | 0.066 |
| LRCN (Donahue et al., 2015)* | 0.284 | 0.305 | 0.223 | 0.128 | 0.089 | 0.067 |
| AdaAtt (Lu et al., 2017)* | 0.295 | 0.308 | 0.220 | 0.127 | 0.089 | 0.068 |
| Att2in (Rennie et al., 2017)* | 0.297 | 0.308 | 0.224 | 0.129 | 0.089 | 0.068 |
| CoAtt (Jing et al., 2017)* | 0.277 | 0.369 | 0.455 | 0.288 | 0.205 | 0.154 |
| HRGR (Li et al., 2018)* | 0.343 | 0.322 | 0.438 | 0.298 | 0.208 | 0.151 |
| MRA (Xue et al., 2018)[+] | N\A | 0.366 | **0.464** | **0.358** | **0.270** | **0.195** |
| Vispi | **0.553** | **0.371** | 0.419 | 0.280 | 0.201 | 0.150 |

Table 1: Automatic evaluations on IU dataset. * results from Li et al. (2018). [+] results from Xue et al. (2018).

key disease information can still achieve high BLEU scores nevertheless it provides limited insight for medical image interpretation. Therefore, ROUGE is more suitable than BLEU for evaluating the quality of automatic reports since it measures both precision and recall. Further, CIDEr is more suitable for our purpose than ROUGE and BLEU since it captures the notions of grammaticality, saliency, importance and accuracy (Agrawal et al., 2017). Additionally, CIDEr uses TF-IDF to filter out unimportant common words and weight more on disease keywords. As a result, higher ROUGE and CIDEr scores demonstrate a superior performance of our medical image interpretation system.

**Evaluation of Disease Classification.** Although ROUGE and CIDEr scores are effective in evaluating the consistency of an automatic report to a manual report, none of them, however, are designed for assessing the correctness of medical report annotation in terms of common thoracic diseases. The latter is another key output of a useful image interpretation system. For example, the automatically generated sentence: "no focal airspace consolidation, pleural effusion or pneumothorax" is considered as similar to the manually written sentence: "persistent pneumothorax with small amount of pleural effusion" using both ROUGE and CIDEr scores despite the completely opposite annotations. Therefore, we assess the accuracy in medical report annotation by comparing with TieNet (Wang et al., 2018) in disease classification using Area Under the ROC (AUROC) as the metric. Our result outperforms TieNet's classification module in 7 out of 8 diseases (Table 2, Fig. 3), even though TieNet is trained on the enhanced version of ChestX-ray8 with 3172 more X-rays and 6 more labeled diseases.

We note that many X-ray based disease classification tasks are multi-label multi-class classification problem. Different from multi-class classification problem where classes are one-hot coded thus mutually exclusive, here we attempt to solve multi-label multi-class classification problem where tasks are inherently related. The classification task of each class is learned simultaneously and synergistically with others using a shared feature representation. As such, the performance of a multi-label classification can benefit considerably from more training samples and classes. Clinically speaking, comorbidity does exist in lung

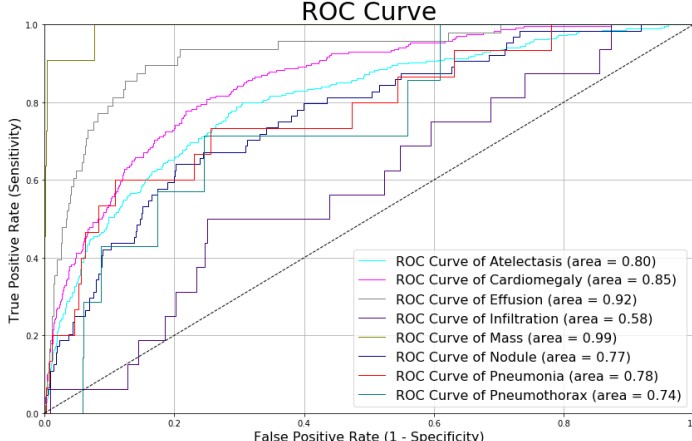

Figure 3: Comparison of disease classification performance using ROC curves.

| Disease | Ate | Cardio | Effusion | Infil | Mass | Nodule | Pneum | Pneumox | **Average** |
|---------|-----|--------|----------|-------|------|--------|-------|---------|-------------|
| TieNet* | 0.744 | 0.847 | 0.899 | **0.718** | 0.823 | 0.658 | 0.731 | 0.709 | 0.757 |
| Vispi | **0.806** | **0.856** | **0.919** | 0.610 | **0.984** | **0.758** | **0.764** | **0.733** | **0.804** |

Table 2: Comparison of disease classification performance using AUROC. * results are from Wang et al. (2018).

diseases, e.g., Infiltration coexists with Effusion and Atelectasis (Wang et al., 2017). Consequently, using additional training samples and extra related disease classes (e.g. 14 classes in (Wang et al., 2018)) can indeed improve the classification performance. Nevertheless, our approach outperforms TieNet with less number of training samples and classes (8 classes in this study). It is likely that TieNet trades image classification performance for report generation performance whereas our model exploits the former to enhance the latter via a bi-level attention mechanism.

**Example System Outputs.** Fig. 4 shows two example outputs each with a generated report and image annotation. The first row presents an annotated "Normal" case whereas the second row presents an annotated "Cardiomegaly" case with the disease localized in a red bounding box on the heatmap generated from our classification and localization module. The results show that our medical interpretation system is capable of diagnosing thoracic diseases, highlighting the key findings in X-rays with heatmaps and generating well-structured reports.

## 4. Conclusions

In summary, we propose a bi-level attention mechanism for automatic X-ray image interpretation. Using only a single frontal view chest X-ray, our system is capable of accurately annotating X-ray images and generating quality reports. Our system also provides visual

| Sample Case | Annotation | Manual Report | Automatic Report |
|---|---|---|---|
| | Normal | the heart size and cardiomediastinal silhouette are within normal limits. pulmonary vasculature appears normal. There is no focal air space consolidation.no pleural effusion or pneumothorax. | the cardiomediastinal silhouette and pulmonary vasculature are within normal limits in size. the lungs are clear of focal airspace disease pneumothorax or pleural effusion. there are no acute bony findings. no acute cardiopulmonary findings. |
| | Cardiomegaly | mild cardiomegaly. mild unfolding of the thoracic aorta. no focal air space opacity. no pleural effusion or pneumothorax. visualized osseous structures are unremarkable in appearance. otherwise no acute cardiopulmonary abnormalities. | mild cardiomegaly. there is no focal consolidation. no pleural effusion or pneumothorax. there is no focal air space consolidation. no pleural effusion or pneumothorax. degenerative changes of the thoracic spine. no acute cardiopulmonary abnormality. |

Figure 4: Illustration of two cases of example outputs of our system.

supports to assist radiologists in rendering diagnostic decisions. With more quality training data becomes available in the near future, our medical image interpretation system can be improved by: (1) incorporating both frontal and lateral view of X-rays, (2) predicting more disease classes, and (3) using hand labeled bounding boxes as the target of localization. We will also generalize our system by extracting informative features from Electronic Health Record (EHR) data and repeated longitudinal radiology reports to further enhance the performance of our system.

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
