# OpenReview forum: "Vispi: Automatic Visual Perception and Interpretation of Chest X-rays"
_MIDL.io/2020/Conference — MIDL 2020_

### Official Review · AnonReviewer2 · 2020-03-05
**Potentially interesting method**

**Rating:** 3
**Confidence:** 2

**Review:**

In this short paper, the authors present a new fully integrated pipeline to diagnose common thoracic diseases. It first annotates the images by classifying and localizing common thoracic diseases with visual support. It then generates a report using a recurrent neural network model dedicated to natural language processing (specifically a an attentive LSTM model). A relatively high level description of the method is given, as this is a short paper. It however allows to understand how the method works. The method is also shown to perform well compared with Li et al. (2018) and Xue et al. (2018).

The paper reads well and the methodology is relatively clear. The results seem promising. The authors however did not made clear what motivated their technical choices compared with Li et al. (2018) and Xue et al. (2018) and why they believe their method outperforms these methods.

---

### Official Review · AnonReviewer1 · 2020-03-11
**No novelty of method or application**

**Rating:** 1
**Confidence:** 5

**Review:**

The authors present a system for generating radiology reports on chest x-rays and additionally providing disease classification and a heatmap to indicate areas of abnormality.

Unfortunately I do not see novelty either of method or of application in this work.  The results of the system are not well analysed or compared fairly with the literature.  Only a single example of a system output is shown with no discussion of cases where various elements of the system fail or limitations.

Specific comments:
The methods to determine the classification and heatmap are not novel (DenseNet and GradCam), nor are the methods to generate the reports (Xue et al, cited).  It appears that this method is simply a combination of previous works.

There is no assessment of how well the GradCam method works to produce heatmaps - it seems to me that producing an accurate heatmap is likely to be one of the most difficult elements of the system and since the authors claim disease localization as one of their contributions it should be properly analysed.

Both table 1 and table 2 show results indicating that this method outperforms others from literature (firstly in report generation, secondly in disease classification).  However in all cases the comparison is unfair since the data used for testing is not the same.  Not only the test set is not the same, but in particular the authors have filtered out images/reports that are "non-relevant" to the 8 abnormalities they are interested in.  The other literature does not make any mention of this step.  This filtering would make it substantially easier to achieve better results.

In table 2, Wang et al work on a  different (larger) dataset with 14 labels (many of which have some overlap/similarity of appearance) so it is not a fair comparison to simply pick out their results on the 8 labels being analysed in this work.

---

### Official Review · AnonReviewer3 · 2020-03-14
**The method is clearly written and comparison with previous works are complete.**

**Rating:** 3
**Confidence:** 3

**Review:**

Pros:
The proposed method uses CNN for image classification on Chest X-rays and CNN-RNN structure is then applied to generate reports. Different strategy is applied for normal/abnormal cases. For abnormal cases, localized abnormal areas extracted from the first CNN is used for CNN-RNN to generate reports.  The result shows good improvement and many state of the art methods are compared.
Cons:
During the second CNN-RNN step, for abnormal cases, localized abnormal areas and the global image is sent to the network separately, so for the normal global image part, the generated sentence may conflict the first step prediction.

---

### Official Review · AnonReviewer4 · 2020-03-16
**Interesting approach but lack of rigorous experiments**

**Rating:** 3
**Confidence:** 4

**Review:**

This work proposed an approach that generates radiology reports from chest x-ray images by splitting pathology-related sentences and the others. The authors should run a model without the pathology(abnormal) sentence generation and show the results to shed some light on how much gains we actually obtain from doing that. The experimental setup should be more rigorous. Does the training/validation/test split guarantee no patient overlap across the subsets? Do the results of the other methods come from their papers or experiments reproduced by the authors?

---

### Meta-Review · Area_Chair1 · 2020-04-06
**MetaReview of Paper276 by AreaChair1**

**Rating:** 3

**Metareview:**

This paper presents a pipeline for automatic interpretation of  medical imaging and learning-based diagnosis. On the one hand, the paper tackles an important and challenging problem, but on the other hand, the validation is very limited and novelty over previous work is unclear. This is therefore a borderline paper.

**Paper Type:**

both

---

### Decision · Program_Chairs · 2020-04-11

Accept